# High Prevalence of Plasmid-Mediated Quinolone Resistance among ESBL/AmpC-Producing Enterobacterales from Free-Living Birds in Poland

**DOI:** 10.3390/ijms241612804

**Published:** 2023-08-15

**Authors:** Beata Furmanek-Blaszk, Marian Sektas, Bartosz Rybak

**Affiliations:** 1Department of Microbiology, Faculty of Biology, University of Gdansk, Wita Stwosza 59, 80-308 Gdansk, Poland; marian.sektas@ug.edu.pl; 2Department of Environmental Toxicology, Faculty of Health Sciences with Institute of Maritime and Tropical Medicine, Medical University of Gdansk, Debowa Str. 23A, 80-204 Gdansk, Poland; bartosz.rybak@gumed.edu.pl

**Keywords:** *Enterobacterales*, free-living birds, ESBL/AmpC, PMQR, plasmid

## Abstract

In this study, we investigated the occurrence of plasmid-mediated quinolone resistance (PMQR) in extended-spectrum β-lactamase- (ESBL) and/or AmpC-type β-lactamase-producing *Enterobacterales* isolates from free-living birds in Poland. The prevalence of the *qnrB19* gene was 63%, and the distribution of isolates in terms of bacterial species was as follows: 67% (22/33) corresponded to *Escherichia coli*, 83% (5/6) to *Rahnella aquatilis*, 44% (4/9) to *Enterobacter cloacae* and 33% (1/3) to *Klebsiella pneumoniae*. The *qnrB19* gene was also found in a single isolate of *Citrobacter freundii*. The molecular characteristics of *qnrB19*-positive isolates pointed to extended-spectrum beta lactamase CTX-M as the most prevalent one (89%) followed by TEM (47%), AmpC (37%) and SHV (16%). This study demonstrates the widespread occurrence of PMQR-positive and ESBL/AmpC-producing *Enterobacterales* isolates in fecal samples from wild birds. In this work, plasmid pAM1 isolated from *Escherichia coli* strain SN25556 was completely sequenced. This plasmid is 3191 nucleotides long and carries the *qnrB19* gene, which mediates decreased susceptibility to quinolones. It shares extensive homology with other previously described small *qnrB19*-harboring plasmids. The nucleotide sequence of pAM1 showed a variable region flanked by an oriT locus and a Xer recombination site. The presence of a putative recombination site was detected, suggesting that interplasmid recombination events might have played a role in the development of pAM1. Our results highlight the broad geographical spread of ColE-type Qnr resistance plasmids in clinical and environmental isolates of *Enterobacterales*. As expected from the results of phenotypic susceptibility testing, no resistance genes other than *qnrB19* were identified.

## 1. Introduction

Fluoroquinolones are commonly used in antimicrobial treatment in both human and veterinary medicine worldwide because of their pharmacodynamic and pharmacokinetic properties [1]. As a consequence of their intense use, the quinolone resistance rate has increased considerably over the last few years. The main mechanism of quinolone resistance is the accumulation of mutations in the genes encoding the subunits of the drug’s target enzymes: DNA gyrase and DNA topoisomerase. In addition, enhanced efflux pump activity or decreased levels of expression of porins have been described [1]. Moreover in recent years, plasmid-mediated quinolone resistance (PMQR) has also been reported [1]. Three transferable quinolone resistance mechanisms have been identified to date: (1) the Qnr proteins of the pentapeptide repeat family that protect DNA gyrase and topoisomerase IV from quinolones, (2) acetylation of quinolones and (3) active efflux pumps that confer only low-level resistance to quinolones [1,2,3]. These PMQR determinants can be spread horizontally among different bacterial species [4]. Genes for PMQR have been found on plasmids varying in size and incompatibility specificity [4]. The low resistance allows the bacterial population to reach a concentration at which secondary mutations to higher resistance can occur [5].

Plasmid-mediated quinolone resistance may facilitate the spread and increased frequency of quinolone-resistant strains. So far, six different *qnr* genes (*qnrA*, *qnrB*, *qnrC*, *qnrD*, *qnrS*, and *qnrVC)* have been discovered, with a number of alleles showing minor variations in nucleotide sequences [4]. These genes are usually plasmid-mediated and can easily spread among the members of *Enterobacterales* through gene transfer mechanisms [2]. Of the known *qnr* families, *qnrB*, has been found worldwide in different pathogens; however, almost two-thirds of the *qnrB* alleles were discovered in the *Citrobacter* genus [6]. The localization of determinants on plasmids is associated with a risk of a dissemination and selection of resistance. Until now, *qnr* genes have been widely detected in different parts of the world and have been found on plasmids carried by environmental organisms [7,8]. The recent detection of *qnr*-bearing plasmids in water organisms suggests that freshwater in inhabited areas may be a reservoir in which pathogens acquire these elements [7,9]. It is possible that environmental quinolone accumulation has contributed to the success of these genes, perhaps by helping to maintain a reservoir of aquatic organisms for which a low-level quinolone resistance gene provided a survival advantage.

The first plasmid-borne quinolone-resistant gene, *qnrB*, was identified in multiresistance plasmid pMG252 from the *Klebsiella pneumoniae* strain [3]. Since that time, the number of new *qnrB* alleles has been continuously rising [1]. *qnrB19* genes have been reported on plasmids of varying size and incompatibility groups (IncN and IncL/M) and on small ColE-like plasmids in *E. coli*, *Salmonella enterica* and *Klebsiella* spp. [10,11,12,13,14]. Moreover, *qnrB19* has been found located in different genetic environments, such as part of the transposon Tn2012 or of the insertion sequence ISCR1 complex class, flanked by IS 26 elements [15,16,17,18,19]. ColE1 plasmids harbor important resistance determinants in humans, animals and the environment, making them important vehicles for the spread of antibiotic resistance, especially within the *Enterobacterales* order [20].

In Poland, data on the occurrence of *qnrB* alleles in fecal isolates recovered from wild birds are limited. In this study, we evaluate its frequency in epidemiologically unrelated avian isolates collected during the period from February to June 2017. In a previous study from our laboratory, we analyzed 33 *E. coli* strains and found them to be extended-spectrum β-lactamase (ESBL) and/or AmpC-type β-lactamase isolates containing different *bla* gene combinations [21]. We genotypically characterized 19 ESBL/AmpC-producing non-*E. coli Enterobacterales* and investigated the prevalence of the *qnrB19* gene in all (*n* = 52) collected isolates. We also characterized a small ColE-like plasmid encoding QnrB19 carried by the *E. coli* SN25556 strain suspected of ESBL production isolated from the fecal sample of a Eurasian coot (*Fulica atra*). Extensive examination of this plasmid indicates that similar plasmids carrying the *qnrB19* gene circulate both in human and animal ecosystems.

## 2. Results

Several studies have reported PMQR-harboring bacterial strains as major global human pathogens [1,2,3,4,7,22]. Some plasmids isolated from bacteria of both clinical and environmental origin and carrying *qnr* genes have been studied in order to better understand their dissemination mechanisms. In Poland, data on the occurrence of *qnr*-carrying plasmids from environmental isolates are lacking. Therefore, we determined the complete nucleotide sequence of a *qnrB19*-carrying plasmid originating from an *E. coli* SN25556 strain isolated from a wild migratory Eurasian coot.

### 2.1. Complete Nucleotide Sequence of pAM1

When the total cellular DNA of *E. coli* SN25556 was separated by electrophoresis through 0.8% agarose gel, a single extra chromosomal DNA band was observed. To investigate this extra chromosomal element in greater detail, plasmid DNA was purified. The plasmid was subjected to extended restriction analyses in single and double digests. Restriction analysis of the single- and double-digested plasmid revealed a fragment pattern on the basis of which the pAM1 plasmid was assigned ~3.0 kb. When plasmid DNA was digested with EcoRI or HindIII, a distinct band of ~3.0 kb in size appeared. Digestion of this plasmid DNA with BglII yielded two detectable fragments of ~2.5 and 0.6 kb. On the basis of these results, a single plasmid of about 3.1 kb designated pAM1 was identified. The occurrence of pAM1 was not affected by the bacterial growth stage or repeated subculturing. To shed further light on the possible genetic advantage provided by pAM1, the nature of the plasmid was examined. Nucleotide sequence analysis of pAM1 revealed a circular molecule of 3191 bp with a mean GC content of 47.82%, which is lower than that of *E. coli* chromosomal DNA (between 50.4% and 50.8%) [23]. Sequence analyses showed that pAM1 is identical or highly related to other known small *qnrB19*-carrying plasmids isolated from enterobacteria. A search within the NCBI database revealed that pAM1 is almost identical to plasmid pMUB-MIN9-7 (accession number: CP069690.1), which was recently identified in a clinical *E. coli* isolate recovered from a patient at the Medical University of Bialystok, Poland, in March 2018. The comparative analysis revealed that both plasmids are of the same size (3191 bp) and differ by only two nucleotides. The places where the corresponding strains were isolated are located 300 km apart from each other in northern and eastern Poland.

Several other plasmids from the database showed a high identity value. The pAM1 plasmid also shared a high degree of similarity with plasmids pQNR20-MO00078 and pQNR20-ABO1775 (100% coverage and 100% identity), pQNR20-MO00080 (100% coverage and 99.97% identity) and plasmid pSA18578_3 (100% coverage and 100% identity) isolated from *E. coli* and *Salmonella enterica* subsp. *Enterica*, respectively. A comparison of pAM1 with highly homologous pQNR20-MO00078, pQNR20-ABO1775 and pQNR20-MO00080 revealed an additional 92 bp insertion downstream of repeated sequences resembling the iteron structure of theta replication plasmids.

Plasmids exhibiting nucleotide similarities >90 to pAM1 have been recovered from hospitalized patients (accession numbers: CP069690.1, CP065188., CP 69577.1 and CP039609.1), poultry meat and feces (accession numbers: CP081512.1, CP081511.1, CP081524.1 and CP081658.1), food (accession number: CP080516.1), the environment (accession number: CP048315.1) and wastewater treatment plants (accession numbers: AP022153.1 and MF554640.1). In addition, pAM1-like plasmids seem to be broadly distributed. They have been detected in Brazil (CP045445.1), Canada (CP81655.1-81658), France (CP065188.1), Germany (CP0581520.1-581672), Japan (AP022153.1), Russia (CP060511.1), Saudi Arabia (CP080516.1), Switzerland (CP048315.1), Trinidad and Tobago (CP066334.1), the UK (CP057009.1) and the USA (CP082455.1, CP082379.1, CP069577.1, MK191837.1 and MK191842.1). One interesting finding is that the *E. coli* SN25556 strain had highly similar *qnrB19* plasmid to that of *Salmonella* from swine and poultry. This reflects a likely transmission of plasmids between *E*. *coli* and *Salmonella* spp. [13,24].

Analysis of the nucleotide sequence of pAM1 revealed the presence of four putative ORFs of at least 90 amino acid codons in length. Three of them have the same transcription direction, and one was encoded on the opposite strand. To attribute functions to the deduced products of the ORFs, these were compared to the gene products available in the databases. ORF1 and ORF2 encode proteins showing a relatively high amino acid sequence similarity with several plasmid replication initiation proteins in enterobacteria (accession numbers: AGM37763.2, SSW86075.1, SXK93043 and VTN84844). A noteworthy feature of this DNA fragment is *orf1* and *orf2* overlap. DNA sequence analysis revealed that both *orf*s are disrupted by nonsense mutations introducing a stop codon and resulting in the production of prematurely truncated proteins. This indicates that neither *orf1* nor *orf2* is responsible for plasmid replication.

We also noticed that the upstream region of *orf/orf2* shares extensive similarity with the backbone of the ColE1-like plasmids [25]. A consensus sequence for the ColE1 replication origin (*oriV*) was located at positions 534 to 536. Additionally, the pAM1 replication region included putative regions containing RNA transcripts RNAII and RNAI, which control the initiation of DNA replication. We also noticed that the upstream region of *orf1/orf2* has an interesting feature of the theta-type replicons. This stretch of DNA contains a 0.1 kb fragment, from nt 2807 to nt 2902, which includes five copies of 11 bp-long direct repeats separated by an 11 bp spacer from two more 11 bp direct repeats. This set of repeated sequences resembles the iteron structures of theta replication plasmids. In earlier studies, Lin et al. showed that direct repeats do not affect the stability or copy number of ColE1-type plasmids; however, they promote the transcription of both the RNAI and preprimer RNA genes [26].

Further downstream, another ORF was detected, coding for a 226-amino-acid protein indistinguishable from the QnrB19 proteins previously reported in *E. coli*, *Salmonella* spp. and *Klebsiella* spp. located on IncN, IncL/M and ColE-like plasmids. The presence of *qnrB19* genes on structurally diverse plasmids points towards their dissemination and adaptation in new host as a relevant factor in the emergence of transferable quinolone resistance. Although qnr genes have frequently been observed in the company of other resistance mechanisms, the pAM1 plasmid has only a quinolone resistance determinant.

ORF4 encodes a 215-amino-acid protein of unknown functions displaying a striking sequence similarity to the hypothetical protein present in many enterobacteria.

### 2.2. Characterization of the Genetic Environment of qnrB19

Plasmid pAM1 contains a ColE-like backbone and a *qnr* region. The ColE-like backbone includes putative regions for plasmid replication (*oriV*, RNAII and RNAI) and mobilization (oriT region); however, it lacks the *mob* and *tra* genes. The regions flanking the *qnrB19* gene show 100% identity with those present in transposon Tn5387 from a *Klebsiella pneumoniae* clinical isolate from the United States (210 bp upstream and 284 downstream) [27] and Tn2012 from an *Escherichia coli* clinical isolate from Colombia (119 bp upstream and 284 bp downstream) [15].

Upstream of *qnrB19*, we located two imperfect 14 bp IR sequences (six and seven mismatches) with similarity to the inverted repeats of IS*Ecp1* (5′-CCTAGATTCTACGT-3′) (Table 1). The distance that separates the *qnrB19* gene from the IRR sequences varies within pAM1, indicating that different insertion events have occurred. A previous study showed that IS*Ecp1* elements recognize a variety of DNA sequences as IRRs and could mobilize adjacent sequences by transposition [28]. For some IS, for example, as recently shown for IS*26*, a single copy can move adjacent resistance genes [16]. It is well known that IS*Ecp1*-like elements are often located at the 5′ ends of several β-lactamase and *qnr* genes to be transposed to other DNA target sites [29]. Upstream of *qnrB19*, the homology with Tn5387 ends in correspondence with a putative imperfect IRR1 for IS*Ecp1* located within a truncated *pspF* reading frame. The *pspF* gene encodes a transcriptional activator of the *psp* operon, and the truncated gene was previously identified as part of the conserved genetic environment of the *qnrB19*-harboring plasmids. An interesting observation is the detection of a 5 bp A+T-rich motif immediately upstream of IRR1, which was shown to represent a potential target site for IS*Ecp1* (TTATA), suggesting that the IS*Ecp1*-like element in the genetic environment of the *qnrB19* gene is involved in the mobilization.

Downstream of the q*nrB19* gene, we detected the 3′ end of a reading frame (positions 2041-2322) whose deduced amino acid sequence showed 100% identity to the 90 C-terminal amino acids of the hypothetical protein found in many *Citrobacter* strains. A previous study evidenced *Citrobacter* as the origin of *qnrB* genes and considered the chromosome of *Citrobacter* as the likely source of plasmid-mediated *qnrB* [6]. Interestingly, in all *qnr*-positive *Citrobacter* strains, the physical distance between the two loci is constant and equals nine nucleotides.

Plasmid pAM1 was also found to carry the Xer module, located approximately 130 bp downstream from the incomplete reading frame, containing a binding site for the host-encoded recombinase XerCD and the ArgR accessory protein. Conserved A-T tracts phased at approximately 10 bp intervals facilitating the curvature of the region between the ArgR and XerC/XerD binding sites were also observed. For ColE1 resolution, XerC/D proteins act on a specific site called cer, a non-codifying region of 280 bp where site-specific recombination occurs. This process is catalyzed by XerC within a sequence of 30 bp composed of two 11 bp half-sides and a central region of 8 bp. The highly conserved Xer site-specific recombination system participates in plasmid evolution as the mechanism of resolution of cointegrates formed between different plasmids by recombination at the *oriT* sites [19].

### 2.3. Resistance Pattern of E. coli SN25556

Antimicrobial resistance in Gram-negative bacteria poses a major threat to human health around the world. *E. coli* remains one of the most frequent causes of common bacterial infections. Antimicrobial sensitivity studies revealed that *E. coli* SN25556 showed decreased sensitivity to several quinolones (Table 2). PMQR genes confer low-level quinolone resistance that is below the CLSI breakpoint for nonsusceptibility, similar to that conferred by first-step DNA gyrase mutations, transporters that extrude quinolones and decreased levels of expression of porins. This strain displayed sensitivity toward most β-lactam antibiotics. Data regarding antimicrobial resistance for *E. coli* strains have varied. Differences in antimicrobial susceptibility between *E. coli* may be related to the source of the isolates and the frequency of use of antimicrobial agents prescribed for treatment of *Escherichia* infection in a particular geographic area.

### 2.4. Detection of Genes Coding for ESBL Resistance and AmpC Lactamase

A previous study focused on investigating ESBL/AmpC-producing *E. coli* (*n* = 33) [21]. In this work, we aimed to characterize non-*E. coli Enterobacterales* (*n* = 19): *E. cloacae* (*n* = 9), *R. aquatilis* (*n* = 6), *K. pneumoniae* (*n* = 3) and *C. freundii* (*n* = 1). ESBL-producing *Enterobacterales* often possess one of three types of genes: *bla_C_*_TX-M_, *bla*_TEM_ and *bla*_SHV_. Indeed, our study confirmed that the majority of isolated strains (17 out of 19) (89%) possess a gene coding for CTX-M; nine isolates carried the gene encoding the TEM β-lactamase, while the *bla*_SHV_ gene was only identified in three isolates. AmpC β-lactamase was detected in seven (36%) isolates.

### 2.5. Detection of the qnrB19 Gene

PMQR genes are not capable of conferring resistance against the most clinically important quinolones; however, they are often associated with different ESBL families [7]. In order to gain insights into the prevalence of the *qnrB19* gene in collected ESBL/AmpC-producing isolates (*n* = 52) representing a random sample of strains collected from wild birds in an urban area of Poland, the presence of the *qnrB19* gene was investigated. PCR-directed screening of all 52 enterobacterial strains showed that 33 isolates (62%) were positive for the *qnrB19* gene. The vast majority were *E. coli* (*n* = 22), although *qnrB19* was also detected in *E. cloacae (n* = 4), *R. aquatilis* (*n* = 5), *K. pneumoniae* (*n* = 1) and *C. freundii* (*n* = 1). The specificity of the PCR products was confirmed by nucleotide sequencing of six randomly selected amplicons. Direct DNA sequencing is an expensive and time-consuming technique; therefore, an alternative protocol providing evidence for the presence of the *qnrB19* gene has been developed. A PCR-based restriction fragment length analysis assay allowed for rapid and reproducible detection of PMQR gene *qnrB19*. Amplification products of the expected size (263 bp) obtained for qnrB19-positive strains were digested with restriction endonuclease Csp45I. The results of the PCR- based restriction fragment length analysis are shown in Appendix A. After digestion and separation on a 10% polyacrylamide gel, a characteristic pattern was observed. Consequently, Csp45I digests the amplified 263-bp PCR product to produce two fragments of 129 and 134 bp, respectively.

## 3. Discussion

The inappropriate use of fluoroquinolones and cephalosporins, which are crucial for the treatment of infections caused by Gram-negative bacteria, leads to a significant increase in resistant isolates in humans and animals. Previous studies have shown that PMQR genes are often detected in ESBL-producing *Enterobacterales* isolated from clinical and environmental samples [7]. In particular, PMQR genes, along with ESBL-encoding genes, are likely to be carried by the same plasmid that can be shared with other organisms by horizontal transmission. Indeed, the acquisition of antimicrobial resistance may be related to the achievement of resistance-determinant genes mediated by plasmids, transposons and gene cassettes in integrons [30]. Migratory birds are potential reservoirs for bacterial agents; this may result from exposure to an environment that has been contaminated by infected humans or animals [31]. *Enterobacterales* carrying transferable quinolone resistance plasmid have been reported in different host species and geographic areas [32].

In this study, a *qnrB19*-harboring *E. coli* SN25556 isolate was recovered from a wild bird in a natural environment, indicating that the quinolone plasmid might be maintained in the *E. coli* populations regardless of antibiotic-selective pressure. We also found that this gene was carried by the small ColE-like plasmid pAM1, showing a sequence nearly identical to those of small QnrB19-encoding ColE-like plasmids identified in various *Enterobacterales* widespread in most parts of the world. Extensive examination of pAM1 indicates that similar plasmids carrying the *qnrB19* gene circulate in both human and animal ecosystems. A detailed sequence analysis of plasmid pAM1 suggested that it was most likely derived from an interplasmid recombination event between a *qnrB19*-carrying plasmid and a small mobilizable plasmid [33]. Although qnr genes have frequently been observed in the company of other resistance mechanisms, the pAM1 plasmid has only a quinolone resistance determinant. No IS*Ecp1* elements were detected on pAM1; however, shortened insertion sequences and an inverted repeats region were found. We also identified a genetic element that has features of an IS element but lacks an identifiable transposase. This result is consistent with previously reported epidemiological settings; the species associated with community-acquired infections (*Salmonella* spp. and *E. coli*) mainly harbored the *qnrB19* gene in small ColE1-type plasmids [12]. Similar small QnrB19-encoding ColE-like plasmids have also been identified in various *Salmonella enterica* serovars of human and animal origin in Latin America and Europe [34,35,36]. The homology of pAM1 to known plasmids demonstrates the importance of the sequencing of plasmids, even in well-studied bacteria such as *E. coli*, since new plasmids are still being recovered.

We were interested in the prevalence of the *qnrB19* gene among 52 ESBL/AmpC-positive *Enterobacterales* isolates of avian origin collected between February and June in 2017. A previous study from our laboratory showed that the majority of ESBL/AmpC-producing *E. coli* isolates from wild birds possess genes coding for CTX-M (88%, *n* = 29) and TEM (76%, *n* = 25), followed by SHV (18%, *n* = 6) and AmpC (23%, *n* = 8) [21]. In the current study, we determined the prevalence of ESBL/AmpC genes among non-*E. coli Enterobacterales* (*n* = 19). Generally, it was found that 63% (*n* = 12) of the analyzed strains harbored at least two of the assessed β-lactamase resistance genes. Our data demonstrate that *bla*_CTX-M_ was the most prevalent ESBL gene (89%), followed by *bla*_TEM_ (47%), *bla*_AmpC_ (37%) and *bla*_SHV_ (16%). This is much higher than previously reported in studies performed in the Czech Republic, France, Germany, Italy, Poland, Serbia, Spain and Switzerland [37,38]. Dominance of the *bla*_CTX-M_ group is consistent with the majority of published studies on wild avian species to date [39,40,41,42,43,44]. Interestingly the levels of ESBL-positive samples varied significantly between countries, with more widespread resistance in the Mediterranean region compared to northern Europe [44,45]. It should be noted that no resistance determinants toward carbapenems were noticed among tested strains, and so far in Poland, carbapenemase-producing *Enterobacterales* have not undergone wide dissemination, unlike that observed in Germany [46] and France [47].

ESBL genes have often been reported as being coassociated with genes that encode PMQR. This correlation is particularly significant in clinically relevant bacteria [48], where the co-occurrence of genes conferring resistance against the two most commonly used group of antimicrobials worldwide poses a risk to human health. The prevalence of PMQR genes among ESBL/AmpC-producing *Enterobacterales* has been investigated in different countries across the world [22,49,50]. In Poland, only a few studies have focused on the coexistence of PMQR determinants and β-lactamase-encoding genes among enterobacterial species. Moreover, this phenomenon was identified mainly in clinical isolates [51,52]. The coexistence of ESBL and PMQR genes among *E. coli* strains isolated from a wastewater treatment plant and red foxes was also recently reported by Osinska A et al. [53] and Osinska M et al. [54]. Except for one investigation of 70 *E. coli* isolates reporting four positives for *qnrS1* and *qnrB19* (*n* = 1) [55], no other study has characterized strains collected from wild birds for PMQR determinants.

In our study, we investigated ESBL/AmpC-producing *Enterobacterales* and revealed a significant association between *qnrB19* and *bla* genes. In addition, it is noteworthy that *E. coli* was the major bacterial species among ESBL/AmpC-producing and PMQR-positive *Enterobacterales*—a finding similar to global reports. Notably, almost two-thirds of isolates (63%, 33/52) harbored the *qnrB19* gene. Twenty-two (67%) *E. coli* isolates were positive for the *qnrB19* gene, showing a frequent occurrence of *qnrB19*-positive *E. coli* in Poland. The *qnrB19* gene was also identified in other members of *Enterobacterales*, such as *E. cloacae* (4/9), *R. aquatilis* (5/6), *K. pneumoniae* (1/3) and in a single isolate of *C. freundii*. The presence of the *qnrB19* gene in ESBL/AmpC *Enterobacterales* isolates was associated with various β-lactamase-encoding genes. However, a significantly high rate of coexistence between PMQR resistance gene *qnrB19* and major β-lactam resistance gene *bla*_CTX-M_ was observed. This can be explained by the carriage of both *qnr* and ESBL/AmpC genes in the same plasmids as reported previously [7]. Taking into account the relatively low number of non- *E. coli Enterobacterales*, the incidence of few isolates possessing *qnr* and *bla* genes may reveal only a small part of the problem.

The results obtained from our study are not surprising and confirm the role of birds as reservoirs and potential spreaders of antibiotic resistance strains. Consequently, wild birds might constitute a potential hazard to human and animal health by transmitting multiresistant strains to waterways and other environmental sources via their fecal deposits. Wild birds, particularly migratory waterfowl, can travel immense distances, inhabit a wide variety of environments and may consequently have a significant epidemiological role in the dissemination of resistant bacteria and genes.

The high prevalence of ESBL/AmpC- and PMQR-positive *Enterobacterales* of avian origin raises the possibility that the evolution of these bacteria was driven by natural selection in wildlife as opposed to clinical use of antibiotics. The evolution of clinically relevant antibiotic resistance genes in wild animals and the relationship between natural, agricultural and human ecosystems indicate that the use of a One Health approach is critical for understanding the dynamics of antimicrobial resistance between humans and animals.

## 4. Materials and Methods

### 4.1. Fecal Samples and Bacterial Isolates

A collection of 52 ESBL/AmpC-producing *Enterobacterales* strains found among the previously described 241 avian isolates was examined for the presence of the *qnrB19* gene. These strains were obtained while ringing the birds in the Tri-City region (Gdansk, Sopot, Gdynia, northern Poland) from February to June 2017 [21]. Avian species were tested for fecal bacteria using sterile cotton swabs. The samples were placed in Amies transport medium and transported to a laboratory for further processing. The samples were inoculated on MacConkey agar medium supplemented with cefotaxime (2 mg/L) and incubated overnight at 37 °C for 24 h. One lactose-positive colony grown on MacConkey agar with antibiotic was selected per sample. The collected samples were first identified by colony morphology, Gram staining and biochemical methods (lactose using characteristic growth in triple sugar iron agar and urea–indole reaction). Species identification was confirmed by MALDI-TOF MS according to the manufacturer’s instructions. The isolates comprised *E. coli* (*n* = 33), *Enterobacter cloacae* (*n* = 9), *Rahnella aquatilis* (*n* = 6), *Klebsiella pneumoniae* (*n* = 3) and *Citrobacter freundii* (*n* = 1). A total of 33 ESBL/AmpC-producing *E. coli* were characterized in a previous study (Rybak et al., 2022). In this work, we investigated the distribution of genes coding for ESBL resistance and AmpC lactamase among 19 non-*E. coli Enterobacterales* isolates. The *E. coli* SN25556 strain harboring the *qnrB19*-carrying plasmid was recovered in January 2017 from a cloacal swab of a wild migratory Eurasian coot. The susceptibility profile of this strain suggested the presence of a low-level quinolone resistance mechanism according to CSLI interpretative criteria [56].

### 4.2. Antimicrobial Susceptibility Testing

The antimicrobial susceptibility of the *E. coli* SN25556 strain was assessed by the agar disk diffusion method following the recommendations of the Clinical Laboratory Standards Institute (CLSI) [56]. Twenty antimicrobial drugs (Graso, Poland) were tested: ampicillin (10 µg), amoxicillin/clavulanic acid (20/10 µg), piperacillin/tazobactam (100/10 µg), cefuroxime (30 µg), cefotaxime (30 µg), ceftazidime (30 µg), cefepime (30 µg), meropenem (10 µg), imipenem (10 µg), nalidixic acid (30 µg), ciprofloxacin (5 µg), levofloxacin (5 µg), amikacin (30 µg), netilmicin (30 µg), norfloxacin (10 µg), tetracycline (30 µg), gentamycin (10 µg), tigecycline (15 µg), trimethoprim/sulfamethoxazole (1.25/23.75 µg) and nitrofurantoine (300 µg). After overnight incubation at 37 °C, inhibition zone diameters were measured and interpreted as resistant, sensitive or intermediately sensitive. *E. coli* ATCC 25922 was used as the quality control strain.

### 4.3. Sequencing Strategy

The plasmid DNA was purified from overnight cultures grown in Luria–Bretani broth using a Plasmid Mini Kit (A&A Biotechnology, Gdansk, Poland) according to the manufacturer’s protocol. The concentration of extracted DNA was measured by a NanoDrop ND-100 (Thermo Fisher Scientific, Wilmington, NC, USA). The pAM1 DNA was initially separately digested with different restriction endonucleases to identify suitable fragments that could be generated for cloning purposes. When pAM1 was digested with the EcoRI enzyme (Thermo Fisher Scientific, Wilmington, NC, USA), one restriction fragment of approximately 3.1 kb could be observed. This fragment was gel-eluted using a Gel Out Kit (A&A Biotechnology, Gdansk, Poland), followed by ligation into the pBR322 EcoRI-digested cloning vector. The ligation mixture was transformed into *E. coli* TOP10-competent cells (Thermo Fisher Scientific, Wilmington, NC, USA). The transformant containing the desired recombinant plasmid was designated pAM7.5. For initial sequence information, cycle sequencing reactions were carried out using recombinant plasmid and vector-specific universal primers (Table 3). The nucleotide sequence was determined by Genomed (Genomed S.A., Warsaw, Poland) using an ABI PRISM Big Dye Terminator Cycle Sequencing Reaction Kit (Applied Biosystems; Foster City, CA, USA) and an ABI PRISM 310 automated sequencer (Applied Biosystems; Foster City, CA, USA). The sequence information obtained from the pAM7.5 recombinant clone was used to design sequence-specific primers (Table 3). Using these primers, the complete nucleotide sequence of the pAM1 plasmid was determined and deposited in the NCBI GenBank database under accession number OQ787038.

### 4.4. PCR Amplification and DNA Sequencing

A total of 19 ESBL/AmpC-producing *Enterobacterales* (*E. cloacae* (*n* = 9), *R. aquatilis* (*n* = 6), *K. pneumoniae* (*n* = 3) and *C. freundii* (*n* = 1)) found among the previously described 52 isolates were screened for *bla*_CTX-M_, *bla*_TEM_, *bla*_SHV_ and *bla*_AmpC_ genes by PCR. The genomic DNA was purified from overnight cultures grown in Luria–Bretani broth using a Genomic Mini Kit (A&A Biotechnology, Gdansk, Poland). The concentration of extracted DNA was measured as described above. Specific primers and the PCR conditions were previously described in [21]. The presence of the *qnrB19* gene in all the collected samples (*n* = 52) was screened using a PCR approach, with the following thermal cycling profile: 31 cycles consisting of 30 s at 95 °C for denaturation, 30 s at 53 °C for annealing and 30 s at 72 °C for extension, followed by a final extension step at 72 °C for 2 min. PCR reactions were performed using PCR MIX (A&A Biotechnology, Poland) in a MiniCycler^TM^ thermal cycler (MJ Research, Inc., USA). Based on the nucleotide sequence of the target gene, the set of primers was used to amplify the internal 263 bp fragment for the *qnrB19* gene. The obtained PCR products were analyzed on 10% polyacrylamide gel in 1 ×TBE buffer and stained with ethidium bromide as a visualizing agent. A known DNA ladder marker (A&A Biotechnology, Gdansk, Poland) was electrophoresed simultaneously in order to assess the size of the amplification product. Randomly selected amplicons were purified using a Gel Out Kit (A&A Biotechnology, Gdansk, Poland) and analyzed by sequencing (Genomed S.A., Warsaw, Poland). The nucleotide sequences of the primers used for detection of the *qnrB19* gene are presented in Table 4.

### 4.5. Restriction Analysis of PCR Products

To identify PCR amplification products, we developed an alternative method involving cutting DNA by restriction endonuclease. Amplicons generated with the above PCR were digested with restriction enzyme Csp45I (Promega). Enzymatic digestions were performed by incubating 10 μL of the amplification products with 5 U of CspI45 and 2 μL of the 10× CspI45 buffer in a total volume of 20 μL. The reaction mixture was incubated at 37 °C for 2 h and electrophoresed on 10% polyacrylamide gel. The molecular sizes of the obtained fragments were estimated using a 100–1000 marker (A&A Biotechnology, Gdansk, Poland) as a molecular size reference.

## Figures and Tables

**Table 1 ijms-24-12804-t001:** Sequences identified as IRRs for IS*Ecp1*-mediated transposition.

Description of Sequence	Nucleotide Sequence of IRL and IRRs (14 bp) *^a^*	No of Base Pairs Identical to Perfect IRR
IRL of IS*Ecp1* Deduced perfect IRR IRR1 IRR2 IRR3	5′-CCTAGATTCTACGT-3′ 5′-ACGTAGAATCTAGG-3′ 5′-ACGCAGATCCAGCG-3′ 5′-ACCCAGTAATTCAG-3′ 5′-AAGTTGAGATTATA-3′	8 7 7

*^a^* Underlined nucleotides indicate residues identical to the perfect IRR.

**Table 2 ijms-24-12804-t002:** Antimicrobial resistance profile of *E. coli* SN25556.

Antibiotic	*Escherichia coli* SN25556 Zone Diam (mm)	*Escherichia coli* ATCC 25922 Zone Diam (mm)
Ampicillin	24	21
Amoxicillin/clavulanic acid	18	18
Piperacillin/tazobactam	23	29
Cefuroxime	10	30
Cefotaxime	20	30
Ceftazidime	17	25
Cefepime	28	33
Meropenem	28	32
Imipenem	28	29
Nalidixic acid	15	25
Ciprofloxacin	23	32
Levofloxacin	19	31
Norfloxacin	22	31
Amikacin	21	22
Netilmicin	20	22
Gentamycin	20	28
Tetracycline	22	30
Tigecycline	20	24
Trimethoprim/sulfamethoxazole	25	38
Nitrofurantoine	21	25

**Table 3 ijms-24-12804-t003:** List of primers used for pAM1 sequencing.

Primers	Nucleotide Sequences 5′ to 3′	Reference
pBRforEco pBRrevEco pAM1 pAM2 pAMfor pAMrev	AATAGGCGTATCACGAGGC TAAAGCTTATCGATGATAA CCGCATCAACGACATGCC CTCCTTCCTGGTATCTCCC CGTGAATTCAGCAGCAGCAAG CTGGAATTCACGCAGATCCAGCGT	This study This study This study This study This study This study

**Table 4 ijms-24-12804-t004:** Primers for polymerase chain reaction of the *qnrB19* gene.

Primers	Nucleotide Sequences 5′ to 3′	Size	Reference
qnrB19-F qnrB19-R	GGMATHGAAATTCGCCACTG ^a^ TTTGCYGYYCGCCAGTCGAA ^a^	263 bp	[57]

^a^ M = A or C; H = A, C or T; Y = C or T.

## Data Availability

The data presented in this study are available upon request from the corresponding author.

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
