# Peer review of "High Prevalence of Plasmid-Mediated Quinolone Resistance among ESBL/AmpC-Producing Enterobacterales from Free-Living Birds in Poland"

_ijms, 2023, doi:10.3390/ijms241612804_

Round 1
Reviewer 1 Report
Furmanek-Blaszk et al. have reported on molecular quinolone resistance determinants in Enterobacterales isolated from Polish birds. The work is a puzzle piece contributing to the network of epidemiological knowledge on this subject. I only have minor comments.
1.) General comment: It is recommended that the authors comply with the up-to-date definition of Enterobacterales in contrast to Enterobacteriaceae. If they don’ want to do so, this at least requires explanation in the text.
2.) Abstract, last-but-one sentence: The statement should be weakened. It is true that highly similar mobile genetic element sequences like the one described by the authors have been described from global clinical and environmental isolates. The representativeness of this isolated finding remains, however, uncertain.
3.) Materials and methods, sub-heading “antimicrobial susceptibility testing”: The authors should make clear which version of the CLSI interpretation standard has been used.
4.) Materials and methods, sub-heading “sequencing strategy”: The provided details are insufficient. The methods should be provided in such details that other working groups can repeat the experiments. If the authors feel that the methods chapter is not the best place for this, please provide the required details as appendix data.
5.) Materials and methods, sub-heading “PCR amplification and DNA sequencing”: Again, please provide more details. E.g., describe the precise master mix composition, the applied devices, etc.
Author Response
Author's Reply to the Review Report (Reviewer 1)
1.) General comment: It is recommended that the authors comply with the up-to-date definition of Enterobacterales in contrast to Enterobacteriaceae. If they don’t want to do so, this at least requires explanation in the text.
The reviewer’s recommendation has been followed – Enterobacterales has now been used in all cases.
2.) Abstract, last-but-one sentence: The statement should be weakened. It is true that highly similar mobile genetic element sequences like the one described by the authors have been described from global clinical and environmental isolates. The representativeness of this isolated finding remains, however, uncertain.
The statement has been weakened in accordance with the reviewer’s suggestion – it now reads:
Our results highlight the broad geographical spread of ColE-type Qnr-resistance plasmids in clinical and environmental isolates of Enterobacterales.
3.) Materials and methods, sub-heading “antimicrobial susceptibility testing”: The authors should make clear which version of the CLSI interpretation standard has been used.
This has been done. The 30th ed. CLSI supplement M100 version of the CLSI interpretation standard has been used and this is now indicated in the references [56]
4.) Materials and methods, sub-heading “sequencing strategy”: The provided details are insufficient. The methods should be provided in such details that other working groups can repeat the experiments. If the authors feel that the methods chapter is not the best place for this, please provide the required details as appendix data.
Fuller details of the methods used have been inserted into the text so that other working groups should now be able to repeat the experiments.
5.) Materials and methods, sub-heading “PCR amplification and DNA sequencing”: Again, please provide more details. E.g., describe the precise master mix composition, the applied devices, etc.
Fuller details and other information has been given to make the methodology more clear and repeatable.
Reviewer 2 Report
The manuscript entitled “High Prevalence of Plasmid Mediated Quinolone Resistance among ESBL/AmpC-Producing Enterobacteriaceae from Free Living Birds in Poland” highlights a good structure and organization of the survey well conducted by the authors.
The topic addressed is topical and of considerable importance to those involved in this issue and to the scientific community in general.
The introductory section adequately approaches the subject and is accompanied by sufficient and up-to-date bibliography.
The results are laid out clearly, in detail, and are aptly divided into the various paragraphs. However, I consider the photograph shown by the authors in Figure 1 unnecessary, which they may possibly keep as supplementary material.
The determination of antibiotic susceptibility would have been appropriate to conduct it by MIC on microtiter plates, but the method employed by the Authors is also sufficient, but they could add a table in which they report the values of this test.
With the exception of these minor considerations, which, however, could improve the manuscript in my opinion, I believe that the authors have done a good job, congratulate them on their results, and look forward to the speedy publication of their survey.
Author Response
Author's Reply to the Review Report (Reviewer 2)
The results are laid out clearly, in detail, and are aptly divided into the various paragraphs. However, I consider the photograph shown by the authors in Figure 1 unnecessary, which they may possibly keep as supplementary material.
As suggested by the reviewer, the photograph, Figure 1, is now given as a supplementary material.
The determination of antibiotic susceptibility would have been appropriate to conduct it by MIC on microtiter plates, but the method employed by the Authors is also sufficient, but they could add a table in which they report the values of this test.
The values of the test conducted are now presented in table form – Table 2.